# Supply Chain Integration, Interfirm Value Co-Creation and Firm Performance Nexus in Ghanaian SMEs: Mediating Roles of Stakeholder Pressure and Innovation Capability

Hongyun Tian [1,*] , Samuel Kofi Otchere [1,*] , Cephas P. K. Coffie [2,3] , Isaac Adjei Mensah [4] and Raphael Kwame Baku [5]

1    School of Management, Jiangsu University, 301 Xuefu Road, Zhenjiang 212013, China
2    School of Management and Economics, University of Electronic Science and Technology of China, Chengdu 611731, China; coffiecephas@gmail.com
3    All Nations University Business School, All Nations University, KF 1908 Koforidua, Ghana
4    Institute of Applied System Analysis (IASA), School of Mathematics, Jiangsu University, Zhenjiang 212013, China; isaacadjeimensahphd@outlook.com
5    School of Economics and Finance, Jiangsu University, 301 Xuefu Road, Zhenjiang 212013, China; raphaelbaku@yahoo.com
*    Correspondence: 100003511@ujs.edu.cn (H.T.); samme20142015@gmail.com (S.K.O.); Tel.: +86-159-5285-9235 (H.T.); +86-188-5289-9847 (S.K.O.)

**Abstract:** Strategic decisions like supply chain integration and interfirm value co-creation are significant to SMEs' performance. Therefore, this paper aims to find out the relationships between supply chain integration, interfirm value co-creation, and firm performance in Ghanaian SMEs. We employed a structural equation model (SEM) to estimate the responses of 473 SMEs registered with the Association of Ghanaian Industries (AGI) to find the nexus between supply chain integration, interfirm value co-creation, and the performance of Ghanaian SMEs. Further, we test for the mediating role of innovation capability and stakeholder pressure in the relationships between supply chain integration and firm performance and the relationship between supply chain integration and interfirm value co-creation, respectively. We found a positive significant relationship between the variables. Innovation capability mediates the positive relationship between supply chain integration and firm performance. Interfirm value co-creation has a negative relationship with the innovation capabilities of SMEs. Therefore, Ghanaian SMEs can invest in technologies, which promote collaborations with external parties to create value while minimizing cost.

**Keywords:** supply chain integration; interfirm value co-creation; stakeholder pressure; innovation capability; SMEs; firm performance

## 1. Introduction

Globally, businesses face the challenge of delivering cutting edge products and services to satisfy the ever-evolving needs of customers. However, achieving this depends on the successful acquisition, processing, storage, and generation of knowledge based on internal and external business environments [1]. Integrative supply chain integration has emerged as the perfect mechanism for businesses to achieve this agenda strategically [2,3]. It establishes a one-stop link between the information and communication systems of heterogeneous parties to promote continuous information exchanges [4]. Empirically, studies have established a strong positive correlation between supply chain integration and improved firm performance [2,4]. Despite these proofs, Ghanaian SMEs are unable to wholly benefit from supply chain integration because of low information technology diffusion, managerial inefficiencies, and inadequate financial resources [5]. Therefore, there is an urgent need to justify the need for Ghanaian SMEs to actively engage in supply chain integration to realize the far-reaching benefit.

Besides supply chain integration, co-creation has become a promising business strategy promoting the participation of other parties in the design and creation of products and services [6–8]. This collaboration can be business-to-customer or business-to-business. While the former focuses on the contribution of customers, the latter engages other businesses in an interfirm value co-creation [9]. Interfirm value co-creation enables collaborating firms to create value together while sharing expertise and resources. Empirically, interfirm value co-creation is found to positively affect firm performance through the provision of shared-value [10–12]. Further, firms engaged in supply chain integration are likely to engage in interfirm value co-creation. Therefore, the constraints hindering Ghanaian SMEs from actively engaging in supply chain integration could affect interfirm value co-creation strategy deployment too [13]. Again, the authoritarian traits of most SME owners (unilateral decision making), the fear of losing trade secrets, and rivalry affect the probability of SMEs engaging in interfirm value co-creation in Ghana [6]. Consequently, there is a need to establish an empirical nexus between supply chain integration, interfirm value co-creation, and firm performance in Ghanaian SMEs to boost participation in these strategies.

Ghana is home to a dozen SMEs providing varied products and services. Statistically, 85% of all the businesses in Ghana are classified as SMEs. This suggests that economically, this group of businesses contributes to employment, revenue generation, and innovation. Over the past decade, the government of Ghana has instituted reforms to increase the output of SMEs. These reforms encourage stress-free start-ups, the diffusion of information technology, the provision of training and support, and financial assistance. While we expect these changes to significantly affect SMEs' participation in supply chain integration and interfirm value co-creation to boost firm performance, other constraints limiting Ghanaian SMEs create uncertainty requiring research inquisition. However, existing literature [6,13,14] fails to examine this nexus fully in Ghana. We close this gap by examining the relationship between supply chain integration, interfirm value co-creation, and firm performance. Further, we examine the mediating roles of innovation capability and stakeholder pressure in the relationships between supply chain integration and firm performance and supply chain integration and interfirm value co-creation, respectively. We answer the research questions: what are the relationships between supply chain integration, interfirm value co-creation, and firm performance in Ghanaian SMEs? Do innovation capability and stakeholder pressure mediate the relationship between supply chain integration and firm performance, and supply chain integration and interfirm value co-creation? We test for the mediating role of stakeholder pressure and innovation capability, because the former influences the decision making of management, while the latter provides infrastructural supports for supply chain integration and interfirm value co-creation activities.

To estimate these relationships, we employ survey data from 473 SMEs in a structural equation model. The results from the study reveal that there is a significant positive relationship between supply chain integration and firm performance, supply chain integration and interfirm value co-creation, interfirm value co-creation, and firm performance. Further, innovation capability mediates the positive relationship between supply chain integration and firm performance, while stakeholder pressure mediates the positive relationship between supply chain integration and interfirm value co-creation. Consequently, this outcome is significant to SMEs owners, government, and scholars, because it justifies the need for investment in information technology to support these business strategies. It also challenges the government to continually provide policies and management support for SMEs to resolve managerial inefficiencies that hinder the diffusion of these strategies. Further, it creates rich knowledge sources and challenges deeper research inquisition to boost productivity in this sector.

The remainder of the paper is organized as follows: literature review and formulation of hypothesis, explanation of the adopted research methods for the conduct of the study and model development, data analysis, presentation, and the discussion of the results from the analysis and the provision of suitable recommendation for policymakers and future studies.

## 2. Literature Review and Hypothesis Development

This section presents and discusses literature related to the research constructs. Further, the propositions of the study are discussed.

### 2.1. Supply Chain Integration

The deliberate interconnection between supply chain partners to share valuable information on the market, products, customers, and new potential markets to make strategic decisions is supply chain integration [15,16]. Companies with efficient internal integration can successfully create a conducive environment to support supply chain integration with partners to improve organizational performance, because the great internal structure, culture, and processes serve as the solid foundation for any successful external collaboration [17]. Studies point to the positive role of supply chain integration in enhancing organizational performances. Zhao et al. [18] studied 195 Chinese organizations and results from the study reveals the strategic alignment of organizations' competitive advantage with partners in a supply chain integration is key in improving the financial performance. Notably, an organization engaged in supply chain integration can significantly improve performance through enhanced customer service, internal operations efficiency, demand flexibility, new market development, and product development improvements [2]. However, studies assert that for an organization to fully benefit from supply chain integration, the relationship between partners should be aligned properly with the organization's competitive strategy [3,4]. To ensure sustainability in collaboration, sometimes it is in the best interest of organizations to enter partnerships with the win–win strategy to help simultaneously support each organization's objective and to reap the full potential benefits of the collaboration, since each party is motivated towards a common outcome [19]. Irrespective of the benefits accruing from the supply chain integration partnerships, there are potential challenges that could mar the success of these partnerships if not managed properly. Communication failure, inadequate resources, organizational culture and structure, issues of trust, misalignment of strategic objectives, and unresolved conflicts within an organization leads to challenges in a supply chain integration partnership [20]. While these challenges could be linked directly to an organization's own internal structure inefficiencies, there are potential looming external challenges too. Most importantly, the risk of losing trade secrets or giving out sensitive information to key competitors when not intended also become a major challenge as the other challenges mentioned previously [21,22]. Nevertheless, the benefits of collaborations are enticing enough to challenge management towards finding a solution to these potential impending challenges.

### 2.2. Interfirm Value Co-Creation

Value co-creation is a deliberate act to create value in terms of products, services, or processes by engaging actors within a specific ecosystem [23]. It is a manner of business strategy that focuses on the generation and ongoing realization of mutual organization-customer value [24]. This is closely related to supply chain integration, because in both cases, a firm needs to open up to external parties [25]. While these two strategies have varying end-products, we expect SMEs engaged in supply chain integration to participate in interfirm value co-creation. This is because supply chain integration creates a platform for interfirm value co-creation to thrive without extra cost [9]. Nonetheless, these strategic decisions could be driven by stakeholder pressure or the innovative capabilities of the firm [6,14]. In a typical value co-creation ecosystem, members can take different forms: the idea generator, the designer, or an intermediary to create value for all parties involved [26]. Similar to supply chain integration, the issue of transparency is coming up, and members within the partnership are required to clearly define their roles, their identity, and their potential contribution to the partnership [7]. Value co-creation comes in two main categories: business-to-customer platforms, where co-creation activities mainly involve the business and both existing and potential customers; and business-to–business, where businesses within the same industry or different industry co-create value for their mutual

gains [25]. Early studies paid considerable attention to business-to-customer platforms, making studies on business-to-business value co-creation relatively understudied [7]. Previously, organizations contested for resources, customers, and market share with little concern for collaborative creations. However, the current trend favors healthy competition and collaboration within an ecosystem with the aim of sustainable growth [23]. Organizations need to align promptly in a value co-creation partnership to take advantage of the integration of the resources to benefit from the partnership [8,26]. Value co-creation is proven to affect organizations' performance in areas like cost savings, efficient use of resources, and efficient marketing in terms of complementary services.

### 2.3. Firm Performance

Firm performance broadly covers both financial and non-financial aspects. The survival and sustainability of all businesses irrespective of size, location, and nature of service depend on consistent firm performance [13,27]. Yu et al. [28] explain financial performance to include improvement in the cost of production, the collection, and processing of information, while the non-financial performance of firms includes improvement in relationships with partners. Therefore, strategic decisions like supply chain integration and interfirm value co-creation, which lead to improvements in these areas, are desirable for organizational performance [21]. Consequently, we expect Ghanaian SMEs to practice following these strategies to increase firm performance financially and non-financially. An effective supply chain management creates short-term economic benefits and long-term competitive advantages [17]. Although studies suggest a positive relationship between value co-creation and firm performance, few have fully conceptualized it [16]. For example, Rajapathirana and Hui [12] found that supply chain integration can establish a close link between manufacturing and distribution processes to deliver products and services in a timely and efficient manner. Efficient supply chain integration enables manufacturers to use accurate information about customer needs and preferences to accelerate product delivery processes, improve production schedules, and reduce inventory obsolescence [2]. Supply chain integration with suppliers and customers can enhance manufacturers' new product development capability; promote product quality, flexibility, and innovation; and product competitive advantage [21,29]. Enterprise performance refers to the extent to which an enterprise can achieve sustainable competitive advantage by utilizing valuable, rare, imitable, and non-strategic resources. To this effect, the innovation capability of firms plays a significant role in the extent of resource utilization [11]. Therefore, we expect SMEs with high innovative capabilities to improve performance significantly.

### 2.4. Stakeholders Pressure

Recent literature points to the positive significant contribution of stakeholder pressure in improving the overall performance of firms [28]. Stakeholders are individuals or institutions with a direct or indirect interest in the outcomes of an organization. While some stakeholders have a monetary interest in organizations, others seek to protect the interest of the society and environment. Konadu et al. [14] consider ten different external stakeholders: customers, suppliers, competitors, industry associations, local communities, environmental organizations, regulators/legislators, media, and shareholders' funds. While the former has a greater influence on organizations in terms of decision making, the latter has relatively minimal influence except in rare instances [30]. The pressure of stakeholders comes from consumers, investors, regulatory requirements, and even NGOs sometimes. As pointed out in stakeholder theories, primary stakeholders are directly or indirectly involved in the shaping of the organization's goals, which leads to profitability and eventual survival of the business [28]. Pressure from this class of stakeholders has the potential of influencing the decisions and actions of management towards the practice or non-practice of supply chain integration strategy [30]. Evidence supports the significance of stakeholder pressure in driving the practice of supply chain integration, which potentially leads to the improvement in organizational performance [14]. Yu et al. [28] reported that

companies that consider shareholders or stakeholders (including them in the corporate strategy) will have a more positive relationship between corporate performances. This is consistent with Freeman's [31] original research, which indicates that the pressure from internal or external stakeholders would significantly encourage the company to improve performance. Maas and Schuster [32] also point out that by adopting progressive environmental management, a company will face fewer internal/external conflicts, which leads to improvement in firm performance. The pressure from stakeholders drives management to make decisions that promote short- and long-run organizational growth [14]. Following, Ahinful et al. [33] report the positive impact of stakeholder pressure on corporate performance. Betts et al. [34] find a positive correlation between community stakeholder pressure and corporate performance. Again, Ahinful et al. [33] suggest that enterprises adopt an environmental management system to improve environmental performance. Mensah [27] also established a positive relationship between stakeholder pressure and supply chain integration initiatives of firms.

### 2.5. Innovation Capability

The innovation capability of firms is an overall capability encompassing the ability to absorb, adapt, and implement technologies to improve the processes and outcomes of the organization. Rajapathirana and Hui [12] find that the innovation effort of organizations positively affects innovation capability and firm performance. This suggests that activities such as interfirm value co-creation and supply chain integration could increase the probability of firms diffusing new technologies. This is consistent with the findings of Le and Lei [11], to indicate that knowledge sharing between internal and external organizational parties improves innovation capability. According to Shafi [10], interactions with customers and competitors promote innovation within organizations. Thus, activities that promote these interactions should be encouraged to drive the innovation capabilities of firms. Further, Mendoza-Silva [9] reveals that the cordial relationship between internal and external organizational parties greatly improves the innovation capability of firms through knowledge sharing activities. However, Kim et al. [1] find that management capabilities and external networking promotes innovation within firms. Contrary, insufficient resources and structural rigidity stifles initiative within firms. Further, Donkor et al. [13] show that organizations with strategic goals are likely to innovate to enhance firm performance compared to those without a clearly defined strategy. According to Donbesuur et al. [6], the technological innovation of firms is a key ingredient of organizational productivity. This reveals that innovation capability is key in the nexus between interfirm value co-creation, supply chain integration, and organizational performance.

### 2.6. Hypothesis Development

Figure 1 depicts the projected relationships between the study variables. Considering the topic under discussion, the research conceptual model was constructed following relevant literature and the research objectives. The figure depicts the affiliations between supply chain integration, stakeholder pressure, value co-creation, and firm performance, respectively. In all, ten (10) research hypotheses were derived to verify the relationships between the aforementioned variables.

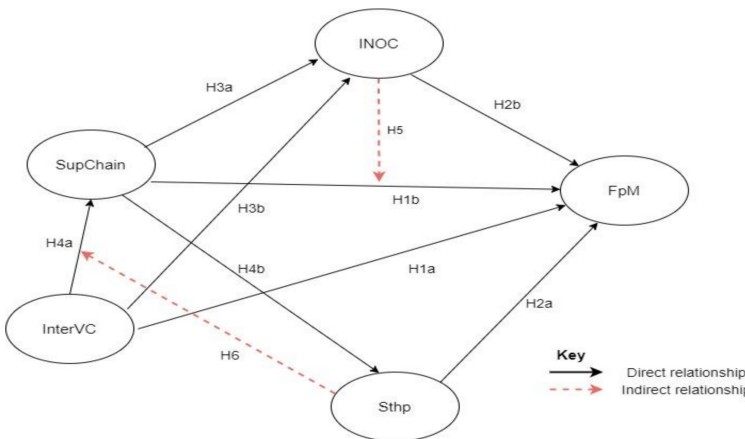

**Figure 1.** Conceptual study model.

Empirically, financial and non-financial firm performance is evidenced to be influenced by interfirm value co-creation and supply chain integration [17,23,25]. Specifically, Frempong et al. [20] establish a positive relationship between new product development, employee creativity, new market entry, and interfirm value co-creation. Thus, SMEs are likely to increase firm performance with interfirm value co-creation. Further, Zhao et al. [18] posit a positive relationship between supply chain integration and efficient resource management, sales growth, and profitability. Consequently, we postulate in hypotheses 1a and 1b that the engagement of Ghanaian SMEs in interfirm value co-creation and supply chain integration improves firm performance.

**Hypothese 1a (H1a).** *Interfirm value co-creation positively affects firm performance in Ghanaian SMEs.*

**Hypothese 1b (H1b).** *Supply chain integration positively affects firm performance in Ghanaian SMEs.*

Besides interfirm value co-creation and supply chain integration, literature establishes a positive nexus between stakeholder pressure, innovation capability of firms, and firm performance [1,12,30,33]. Stakeholder groups have varied requirements, and thus pressure from these heterogeneous groups influences management decisions [34]. According to Mensah [27], stakeholder pressure increases firm profitability and corporate social responsibility. Further, the innovation capability of firms determines how far firms explore resources. Per Rajapathirana and Hui [12], the innovative capability of firms increases efficient resource management and firm profitability. Therefore, we hypothesize 2a and 2b to establish the influence of stakeholder pressure and innovation capability on the performance of Ghanaian SMEs.

**Hypothese 2a (H2a).** *Stakeholder pressure positively affects the performance of Ghanaian SMEs.*
**Hypothese 2b (H2b).** *The innovation capability of Ghanaian SMEs affects firm performance.*

The timely ability of firms to employ new methods, tools, and equipment to execute the tasks is evidenced to be influenced by the openness of organizations to assimilate both internal and external information [4,24]. According to Mendoza-Silva [9], firms engaging in supply chain integration have a higher probability of using new tools and methods. This is possible because improved processes and methods are required for efficient integration. Further, interfirm value co-creation positively influences the innovativeness of firms [10,23]. This is because firms have the opportunity to acquire new skills and use new methods or processes through the interfirm co-creation process. Therefore, we hypothesize 3a and 3b to estimate the influence of supply chain integration and interfirm value co-creation on the innovation capability of Ghanaian SMEs.

**Hypothese 3a (H3a).** *Supply chain integration practice in Ghanaian SMEs positively affects innovation capability.*

**Hypothese 3b (H3b).** *Interfirm value co-creation practice in Ghanaian SMEs positively affect innovation capability.*

Supply chain integration is a strategic decision that requires the commitment of firm resources, whose outcome should benefit all stakeholders. Consequently, literature establishes a positive relationship between stakeholder pressure and the practice of supply chain integration in firms [14,17,27]. Further, to maximize resources, firms engaged in supply chain integration are likely to engage in interfirm value co-creation to share the information acquisition and distribution infrastructure within the organization [4,17,24]. Therefore, we posit hypothesis 4a to establish the influence of supply chain integration on the diffusion of interfirm value co-creation in Ghanaian SMEs. Further, we theorize hypothesis 4b to find the influence of stakeholder pressure on the decision of Ghanaian SMEs to engage in supply chain integration.

**Hypothese 4a (H4a).** *Supply chain integration positively drive the diffusion of interfirm value co-creation in Ghanaian SMEs.*

**Hypothese 4b (H4b).** *Stakeholder pressure positively influences supply chain integration practice in Ghanaian SMEs.*

Empirically, supply chain integration and innovative capability of firms are evidenced to influence firm performance positively [4,6,12,25]. However, these variables exert different influences on firm performance. Further, the firm innovative capability is theorized to influence supply chain integration success and firm performance [12,13]. Thus, because the innovative capability of firms determines how best supply chain integration is executed, the nexus between supply chain integration and firm performance could be explained by innovation capability. Therefore, we assume that the relationship between supply chain integration and SMEs performance in Ghana is influenced by innovation capability.

**Hypothese 5 (H5).** *The innovation capability of Ghanaian SMEs mediates the positive relationship between supply chain integration and firm performance.*

Stakeholders influence the decision-making process or management [27,32]. Therefore, strategic decisions like supply chain integration and interfirm value co-creation could be influenced by stakeholder pressure. Empirically, literature establishes that stakeholders' pressure influences supply chain integration in firms [14,30,32]. Further, Yu et al. [28] find a significant positive relationship between stakeholder pressure and interfirm value co-creation. Thus, the relationship between supply chain integration and interfirm value co-creation could be strengthened by pressure from stakeholders. Consequently, we theorize that stakeholder pressure influences the relationship between supply chain integration and interfirm value co-creation in Ghanaian SMEs.

**Hypothese 6 (H6).** *Stakeholder pressure mediates the positive relationship between supply chain integration and interfirm value co-creation practice in Ghanaian SMEs.*

## 3. Research Methods

### 3.1. Data Source and Sampling Procedure

The study explores the relationship between supply chain integration, interfirm value co-creation, and firm performance. Further, we examine the mediating roles of innovation capability and stakeholder pressure. The study is quantitative in nature with supporting survey data from SMEs in Ghana. Currently, Ghana is home to several SMEs [5,6]. Therefore, to streamline the study, we employ two sampling techniques: purposive non-probability sampling and simple random probability sampling. Using the purposive sampling technique, we include only SMEs documented with the association of Ghanaian Industries (AGI). This group of SMEs is appropriate for the study, because they have frequent training on industry best managerial practices. Per AGI, there are 1600 plus active members scattered across the country. Consequently, we are unable to include all these SMEs in the study. Therefore, we sample further using the simple random probability

technique. Consistent with other studies [35], this provides equal inclusion opportunities for SMEs without bias. However, SMEs with invalid e-mails and unreachable telephone numbers are replaced in the process. Following, a total of 645 SMEs with valid emails and reachable telephone numbers are included as the final sample of the study. Per the one-third sampling rule, the final sample of 645 is representative of the study population. Thus, this provides substantiation for the generalization of our research findings. To collect data, a structured five-part closed-ended survey instrument covering respondent demographics, interfirm value co-creation, innovation capability, supply chain integration, stakeholder pressure, and firm performance is employed. The target respondents are the chief executive officers (CEOs) or owners of SMEs. This decision is justified by the fact that most SMEs in Ghana have owners doubling as CEOs. Therefore, strategic decisions like the engagement in interfirm value co-creation and supply chain integration rest with CEOs or the owners. To ensure the suitability of the survey instrument, we employ the split-half by sending a sample of the instrument to 25 of the selected SMEs. Afterward, changes were made to the instrument to provide clarity and ensure internal consistency. To draft the final questionnaire, the improved instrument is further sent to 15 of the selected SMEs. This process produced a refined instrument free from ambiguity. The approved questionnaire was then sent to the 645 SMEs via email. The entire data collection period lasted between March–August 2020. The final response generated yielded 473 (645). This difference is accounted for by no response due to the fact that some SMEs did not attend to their emails or answer our follow-up telephone calls and the elimination of participants based on incomplete responses. Nonetheless, the response rate is great and thus reduces the probability of non-response rate bias. Therefore, the outcome of the study is not affected by the number of respondents.

### 3.2. Measurement of Constructs

In the context of construct measurements, the questionnaire was structured to evaluate the relationship between five construct variables, which include supply chain integration, innovation capability, stakeholder pressure, value co-creation, and firm performance. Table 1 provides details of the measurement construct and the number of elements. Specifically, the interfirm value co-creation construct consists of two elements, approach and measure. The first element (approach) focuses on the different methods adopted by SMEs to engage in interfirm value co-creation and consists of four items, whereas the second element (measure) captures the benefits and challenges of interfirm value co-creation in the SMEs and also consists of seven items [7,8,26]. Innovation capability constructs, on the other hand, are characterized by four elements (investment in an information system, investment in new machines, new methods, and new processes). Summarily, each of the aforementioned components of innovation capability construct is measured using an item (one item each). [9,12,13]. Moreover, the supply chain integration construct also has four elements, where system coupling and collaboration between SMEs and others, respectively, are measured using two measurement items each, whereas joint decision together with information sharing is measured using four items each [2,4,15]. Concerning the components of supply chain integration, system coupling focuses on the ability for SMEs to connect their systems with the systems of other firms, while joint decision captures the ability of SMEs to engage in group decision making with heterogeneous parties. Information sharing also focuses on the ability of SMEs to share information between departments and other parties in the supply chain, and collaboration captures the different levels of partnerships engaged in by the SMEs. Further, the stakeholder pressure consists of two elements including internal stakeholders (measured using two items), which captures the influence of parties within the SMEs and external stakeholders (characterized by 10 measurement items) centers on parties outside the business environment of the SMEs [27,30,32]. Finally, the firm performance construct consists of two elements: financial firm performance (consisting of three items) captures the profitability and growth in sales revenue, and non-financial firm

performance (measured by five items) focuses on market share growth, customer retention, and improvement in corporate image of SMEs [25,36].

**Table 1.** Questionnaire development and measurement constructs.

| Constructs | Elements | Measures | References |
|---|---|---|---|
| Inter-firm value co-creation | Approaches | 4 questions | [7,8,26] |
| | Measure | 7 questions | |
| Innovation capability | Information system | 1 question | [9,12,13] |
| | New machines | 1 question | |
| | New methods | 1 question | |
| | New processes | 1 question | |
| Supply chain integration | System coupling | 2 questions | [2,4,15] |
| | Joint decision | 4 questions | |
| | Information sharing | 4 questions | |
| | Collaborations | 2 questions | |
| Stakeholders pressure | Internal | 2 questions | [27,30,32] |
| | External | 10 questions | |
| Firm performance | Financial | 3 questions | [25,36] |
| | Non-financial | 5 questions | |

*3.3. Theoretical Model Specification*

The structural equation model (SEM) is employed in this study to examine the mediating role of stakeholder pressure and innovation capability through the effect of supply chain integration and inter-firm value co-creation on firm performance. Using the SmartPLS, we assess the psychometric properties. A structural equation model is a statistical method that allows complex relationships between one or more independent latent variables and one or more dependent latent variables. Summarily, the structural model establishes relationships between the latent variables both response and explanatory. Specifically, the measurement model theoretically consists of the following equations, using standard notations by Bollen [37];

$$
\begin{aligned}
x_1 &= \lambda_1 \xi_1 + \theta_1 & y_1 &= \lambda_3 \eta_1 + \varepsilon_1 \\
x_2 &= \lambda_2 \xi_2 + \theta_2 & y_2 &= \lambda_4 \eta_1 + \varepsilon_2 \\
x_3 &= \lambda_3 \xi_3 + \theta_3 & y_3 &= \lambda_5 \eta_1 + \varepsilon_3
\end{aligned}
\tag{1}
$$

where $x_s$ and $y_s$ are the observed variables for the latent variables, $\xi_1$'s and $\eta_i$'s represents the latent (constructs) variables, $\lambda_i$'s are the factor weights, and $\theta_i$'s and $\varepsilon_i$'s are the residual terms. In matrix form, the measurement model as specified in Equation (1) is expressed as:

$$
\begin{aligned}
X &= \Lambda_x \xi + \theta \\
Y &= \Lambda_y \xi + \varepsilon
\end{aligned}
\tag{2}
$$

Based on the theoretical model specification, an assumption can be made that series of regression equations are to be estimated to assess the causal effects amid the variables employed in the study, which can be found in the structural model as Figure 1 depicts. Thus, regarding the theoretical model formulated in Equation (1) and the structural model from Figure 1, the series of regression equations to be estimated as specified as follows:

$$
\text{InterVC}_i = \beta_0 + \beta_a \text{FpM}_i + \varepsilon_i
\tag{3}
$$

$$
\text{SupChain}_i = \beta_0 + \beta_a \text{FpM}_i + \varepsilon_i
\tag{4}
$$

$$
\text{InterVC}_i = \beta_0 + \beta_b \text{SupChain}_i + \varepsilon_i
\tag{5}
$$

$$SthP_i = \beta_0 + \beta_b SupChain_i + \varepsilon_i \tag{6}$$

$$InC_i = \beta_0 + \beta_a SupChain_i + \varepsilon_i \tag{7}$$

$$ShP_i = \beta_0 + \beta_c FpM_i + \varepsilon_i \tag{8}$$

$$SupChain_i = \beta_0 + \beta_c InC_i + \varepsilon_i \tag{9}$$

$$InterVC_i = \beta_0 + \beta_c InC_i + \varepsilon_i \tag{10}$$

$$SupChain_i = \beta_0 + \beta_d InC_i^* + \beta_a FpM_i + \varepsilon_i \tag{11}$$

$$SupChain_i = \beta_0 + \beta_d SthP_i^* + \beta_f InterVC_i + \varepsilon_i \tag{12}$$

where FpM, SupC, InterVC, InC, and ShP represent firm performance, supply chain integration, value co-creation, innovation capability, and stakeholder pressure, respectively. Notably, InC*, and ShP* serve as mediating variables. Thus, Equations (11) and (12) estimate the mediating effect of innovation capability and stakeholder pressure on the relationship between supply chain integration and firm performance as well as supply chain integration and interfirm value co-creation, respectively. Table 2 gives a summary of the relationships to be estimated.

**Table 2.** Test of the mediated relationship between variables.

| Model | Predictors | Mediator | Criterion |
|---|---|---|---|
| 1 | InterVC ————————————————————————> | | FpM |
| 2 | SupC ————————————————————————> | | FpM |
| 3 | InterVC ————————————————————————> | | SupC |
| 4 | ShP ————————————————————————> | | SupC |
| 5 | InC ————————————————————————> | | SupC |
| 6 | ShP ————————————————————————> | | FpM |
| 7 | SupC ————————————————————————> | | InC |
| 8 | InterVC ————————————————————————> | | InC |
| 9 | SupC ————————————InC———————> | | FpM |
| 10 | SupC ————————————ShP———————> | | InterVC |

Note: FpM, SupC, InterVC, InC, and ShP represent financial performance, supply chain integration, inter-firm value co-creation, innovation capability, and stakeholder pressure, respectively, ————> indicates the direction of causal relationship to be estimated for a specific model.

## 4. Statistical Analysis

### 4.1. Descriptive Analysis

The online survey instrument gathered data from a total of 473 (100%) SMEs in Ghana was analyzed. Table 3 shows the output. Specifically, we focused on the chief executive officers (CEOs) of these businesses. The rationale behind this is that most SMEs in Ghana are owned and controlled by the CEOs. Therefore, the target respondents are suitable for the objectives of the study. Table 4 depicts the results from the descriptive analysis. Male (77.0%) CEOs dominate the SMEs sector in Ghana. The majority of CEOs in Ghana fall between the ages of 26–30 years (48%) and 31–35 years (27.5%). This is positive for the development of the sector, since younger CEOs have a higher probability to learn [5]. Concerning education, 93.9% of the CEOs have tertiary level education. This also increases the ability of the CEOs to acquire new skills to resolve management inefficiencies [13]. However, most of the SMEs in Ghana engages in service-related businesses (70%). This supports the reason why Ghana depends largely on imports to satisfy the needs of the country (Yahaya et al., 2019). Further, most of the SMEs are located in urban areas (76.5%). This is common in developing countries, because urban areas provide good infrastructure unlike rural areas [5]. Finally, most of the SMEs sampled have between 10–50 employees

(72.1%). This reveals the expansion difficulties of SMEs in the country. Consequently, the study skews more toward smaller businesses than medium-size businesses in Ghana.

**Table 3.** Response rate and characteristics.

| Variables | | Frequency | Percentage |
|---|---|---|---|
| Gender | Male | 364 | 77.0% |
| | Female | 109 | 23.0% |
| Age | 18–25 years | 33 | 7.0% |
| | 26–30 years | 227 | 48% |
| | 31–35 years | 130 | 27.5% |
| | 36–40 years | 40 | 8.5% |
| | 41 years or over | 43 | 9.1% |
| Education | Basic | 4 | 0.8% |
| | Secondary | 25 | 5.3% |
| | Tertiary | 444 | 93.9% |
| Nature of Business | Service | 331 | 70.0% |
| | Manufacturing | 50 | 10.6% |
| | Service and manufacturing | 93 | 19.4% |
| Location | Urban | 362 | 76.5% |
| | Rural | 111 | 23.5% |
| Employees | 10–50 | 341 | 72.1% |
| | 51–100 | 43 | 7.2% |
| | 101 or more | 98 | 20.7% |

**Table 4.** Reliability Test.

| Variables | Cronbach's Alpha | Composite Reliability |
|---|---|---|
| InterVC | 0.738 | 0.823 |
| InC | 0.701 | 0.811 |
| SupC | 0.924 | 0.945 |
| ShP | 0.847 | 0.873 |
| FpM | 0.893 | 0.901 |

*4.2. Reliability and Validity Tests*

To ensure the reliability of the research instrument, we employ Cronbach's alpha and composite reliability tests. Table 4 depicts the test results. Specifically, we use Cronbach's alpha to test for internal consistency and item reliability. Per the results from Cronbach's alpha, all our variables have values greater than 0.7 to suggest that the research instrument is capable of measuring the stated constructs. Further, we use the composite reliability test to measure the overall scale reliability of the instrument. The output shows values greater than 0.8 to prove that overall, the research instrument is reliable. Consequently, this provides adequate evidence to support the outcome of the study.

Further, we proceed to test the validity of the research instrument by employing the cross-loading factor and average variance extracted (AVE). Table 5 depicts the results of the cross-loading factor and the AVE. Hypothetically, the cross-loading factor should have values greater than 0.7. Therefore, items that did not load within this scale were eliminated. In total, 15 items were removed: InterVC (AP4, MS1, MS3, MS4), InC (IN1), SupC (DS1, IS3), ShP (SPE1, SPE5, SPE7, SPE8, SPE9, SPE10), and FpM (FP3, NF3). All the remaining items loaded with values greater than 0.7. This proves that the items adequately support the various constructs within the study.

**Table 5.** Cross-loading factor and average variance extracted.

|  | InterVC | InC | SupC | ShP | FpM |
|---|---|---|---|---|---|
| **AP1** | 0.831 | 0.203 | −0.075 | −0.088 | 0.049 |
| **AP2** | 0.840 | 0.050 | −0.070 | 0.311 | 0.052 |
| **AP3** | 0.803 | −0.004 | −0.009 | 0.228 | 0.516 |
| **MS2** | 0.795 | −0.169 | −0.384 | −0.034 | 0.153 |
| **MS5** | 0.796 | −0.279 | −0.364 | 0.223 | 0.047 |
| **MS6** | 0.769 | −0.361 | −0.322 | 0.184 | 0.069 |
| **MS7** | 0.735 | −0.396 | −0.044 | 0.118 | 0.069 |
| **IN2** | 0.225 | 0.725 | 0.264 | 0.480 | 0.226 |
| **IN3** | 0.074 | 0.784 | 0.087 | 0.254 | 0.244 |
| **IN4** | 0.043 | 0.787 | 0.195 | −0.086 | 0.326 |
| **SCI** | 0.441 | 0.292 | 0.736 | −0.406 | 0.360 |
| **SC2** | 0.583 | 0.350 | 0.838 | 0.031 | 0.027 |
| **DS2** | 0.670 | 0.236 | 0.788 | −0.144 | −0.233 |
| **DS3** | 0.750 | 0.172 | 0.766 | −0.105 | 0.060 |
| **DS4** | 0.629 | 0.283 | 0.811 | 0.075 | 0.029 |
| **IS1** | 0.616 | 0.366 | 0.806 | 0.033 | 0.183 |
| **IS2** | 0.716 | 0.307 | 0.836 | −0.286 | −0.178 |
| **IS4** | 0.690 | 0.171 | 0.813 | −0.214 | −0.287 |
| **CL1** | 0.589 | 0.316 | 0.874 | −0.175 | 0.318 |
| **CL2** | 0.701 | 0.340 | 0.875 | −0.331 | −0.052 |
| **SPI1** | 0.599 | −0.255 | −0.161 | 0.767 | 0.034 |
| **SPI2** | 0.412 | 0.040 | .095 | 0.722 | −0.418 |
| **SPE2** | 0.650 | −0.144 | .009 | 0.780 | −0.213 |
| **SPE3** | 0.576 | −0.252 | −0.002 | 0.803 | 0.002 |
| **SPE4** | 0.549 | 0.164 | 0.131 | 0.828 | −0.209 |
| **SPE6** | 0.516 | 0.261 | −0.019 | 0.786 | −0.195 |
| **FP1** | 0.612 | −0.408 | 0.306 | −0.262 | 0.797 |
| **FP2** | 0.629 | −0.361 | 0.383 | −0.196 | 0.863 |
| **NF1** | 0.698 | −0.229 | 0.369 | −0.054 | .736 |
| **NF2** | 0.633 | −0.151 | 0.320 | −0.181 | 0.804 |
| **NF4** | 0.698 | −0.215 | 0.200 | 0.227 | 0.787 |
| **Average variance extracted** | **0.72** | **0.64** | **0.69** | **0.77** | **0.59** |

The output of the AVE, as shown in Table 5, is expected to be greater than 0.5. Consequently, the constructs employed in our study show average variance extracted values 0.72, 0.64, 0.69, 0.77, and 0.59 to prove that the research instrument meets both reliability and validity requirements to warrant further investigation. Therefore, we proceed to estimate our structural equation model outlined in Figure 1.

*4.3. Hypothesis Testing*

After fulfilling both the reliability and validity test, we test the study hypotheses using the smart PLS. Table 6 is the estimation output. At a 5% (0.05) significant level, we find that all the hypotheses (H1–H10) are accepted. However, except for the relationship between InterVC and InC, which is negative, all the other relationships are positive. Specifically, InterVC positively affects FpM. Thus, a unit increase in the InterVC activities improves FpM by 0.16[a]. This is consistent with the findings of existing studies [19,25]. SupC positively affects FpM; an increase in SupC activities increases FpM by 0.14[a]. This is why scholars advocate for the promotion of SupC between firms [17]. Again, we find that InterVC positively affects SupC within SMEs in Ghana. Thus, a unit increase in InterVC improves SupC by 0.20[a]. Empirically, InterVC activities are proven to encourage SupC within firms [24]. The relationship between ShP and SupC per our analysis is positive. This means an increase in ShP increases SMEs SupC activities in Ghana by 0.47[a]. Stakeholders exert pressure on management to make the right decisions for the benefit of all [27,32]. Further, we find that InC has a positive relationship with SupC. Statistically, a unit increase in InC increases SupC by 0.20[a]. This reveals the need for advanced technologies to adequately

support the activities of SupC [9,10]. Again, we find ShP to positively affect FpM in Ghanaian SMEs. This means a unit increase in ShP increases FpM by 0.55[a]. This supports the significance of ShP in the effective and efficient management of businesses [32,34]. InC also shows a positive relationship with FpM in Ghanaian SMEs. InC is proven to be crucial in enhancing the efficiency and productivity of businesses [1]. Consequently, the results provide that an increase in InC increases FpM by 0.04[a]. However, the relationship between InterVC and InC is negative. This is inconsistent with existing literature [13]. An increase in InterVC negatively affects InC in Ghanaian SMEs. Rationally, we expected a positive relationship between these variables. Further, we estimated the mediating roles of InC and ShP on the relationships between SupC and FpM together with SupC and InterVC, respectively. We evidenced that InC positively mediates the relationship between SupC and FpM. This is consistent with the findings of existing studies. However, with a coefficient of 0.07[b], the strength of the relationship witnessed between SupC and FpM at 0.14[a] is reduced. Finally, we find that ShP mediates the relationship between SupC and InterVC. This outcome is synonymous with the findings of existing studies [27,32]. Equally, the strength of this relationship is reduced given the coefficient of 0.02[a] compared to the original relationship with the coefficient of 0.20[a].

**Table 6.** Hypothesis testing.

| Hypothesis | Relationships | Coefficients | *p* Values | Effect | Results |
|:---:|:---:|:---:|:---:|:---:|:---:|
| H1a | InerVC→FpM | 0.161 | 0.003 [a] | Direct | Supported |
| H1b | SupC→FpM | 0.140 | 0.005 [a] | Direct | Supported |
| H2a | InterVC→SupC | 0.204 | 0.000 [a] | Direct | Supported |
| H2b | ShP→SupC | 0.470 | 0.000 [a] | Direct | Supported |
| H3a | InC→SupC | 0.197 | 0.000 [a] | Direct | Supported |
| H3b | ShP→FpM | 0.551 | 0.000 [a] | Direct | Supported |
| H4a | InC→pM | 0.039 | 0.004 [a] | Direct | Supported |
| H4b | InTerVC→InC | −0.007 | 0.000 [a] | Direct | Supported |
| H5 | —SupC Inc→FpM | 0.066 | 0.020 [b] | Indirect | Supported |
| H6 | SupC—ShP→InterVC | 0.020 | 0.007 [a] | Indirect | Supported |

Note: [a] and [b] represent 1% and 5% level of significance, respectively.

To estimate the robustness of the model estimation, we use the outcome of the adjusted R-square as the model goodness of fit test. Figure 2 depicts the output of the goodness of fit test and path coefficients generated by the SmartPLS. Hypothetically, these models show adjusted $R^2$ values of more than 0.5 to prove that the models explain more than 50% of the variations in the relationships estimated. Consequently, the model performs better above a zero model.

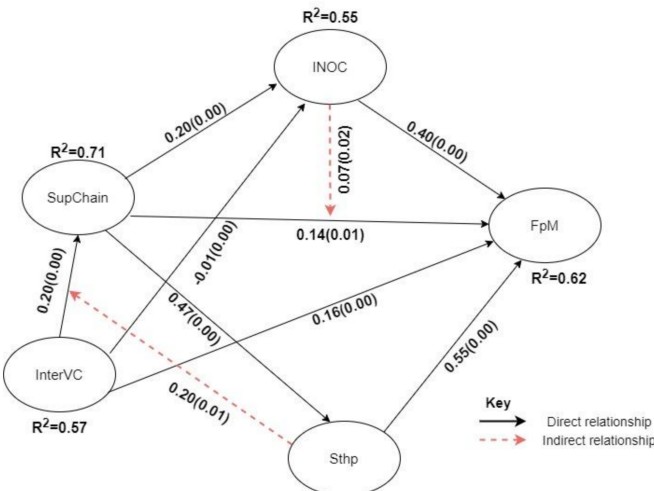

**Figure 2.** Estimation output of the model.

## 5. Discussion

The study estimates the relationships between supply chain integration, interfirm value co-creation, and firm performance in Ghana. Further, we estimate the mediating effects of innovation capability and stakeholder pressure in the relationships between supply chain management and firm performance and interfirm value co-creation and firm performance, respectively. All the hypotheses show a positive relationship except the relationship between interfirm value co-creation (InterVC) and innovation capability (InC).

Synonymous with existing literature [24,25], interfirm value co-creation (InterVC) improves firm performance (FpM) in Ghanaian SMEs. While this is expected, it also signifies that SMEs in Ghana are now more open to collaborations with other businesses, unlike decades ago. Previously, Ghanaian SMEs were known to be secretive about their operations due to the fear of losing trade secrets and customers to their competitors. Therefore, the turnaround in this area could be explained by the current demand for superior goods and heightened competition in the sector [14,27]. Consequently, growth in this sector could be propelled by government policies and incentives that promote collaboration and fair competition in the industry. InterVC positively affects supply chain integration (SupC) in Ghanaian SMEs. This outcome confirms the existing hypothesis on the relationship between SupC and InterVC [16,38]. However, the case of Ghanaian SMEs could be explained by the need to reduce cost and improve efficiency and survival. Currently, the Ghanaian SMEs industry is witnessing expansive growth with the influx of new players. Therefore, the need to create superior goods at the right time and cost is paramount to surviving the competition. SMEs can achieve this with a seamless link between InterVC and SupC to boost productivity. Next, InterVC negatively affects innovation capability (InC) in Ghanaian SMEs. This outcome contradicts existing theories on the relationship between InterVC and InC [14,17]. Therefore, this could be explained by the fact that SMEs in Ghana have minimal innovation capabilities. Again, it could signify the infancy of InterVC activities in Ghanaian SMEs. Consequently, to promote creativity, innovation, and profitability of SMEs in Ghana, InterVC activities should be designed to stimulate innovation.

Stakeholders as a group(s) of individuals with a keen interest in the operations of businesses have been proven to positively influence the successful operations of SMEs [27,30,32]. Consequently, our study finds stakeholder pressure (to ShP) improves SupC in Ghanaian SMEs. This could be explained by the fact that these group(s) of individuals have both monitory and non-monitory interests in the management of the SMEs, and this drives management to make decisions that create overall value for the stakeholders. Decisions motivated by stakeholder pressure could range from engagement in corporate social responsibility, engaging in environmentally friendly resource exploitation, to engaging in SupC [27,33]. Further, we find that InC positively affects FpM in Ghanaian SMEs. Rationally, stakeholders expect higher performance from businesses, and therefore their pressure on management would be directed towards increasing the performance of the firm [28,30]. This provides support to the recent popularity of value co-creation globally [24,25]. Consequently, to drive creativity and productivity amongst SMEs in Ghana, the role of stakeholder pressure is significant. Positive stakeholder engagements should be encouraged to breed an innovative ecosystem.

The seamless planning and execution of information dissemination between parties in a supply chain integration management (SupC) have proven to empirically impact on a firm's performance [19–21]. Similarly, our study finds SupC positively affects FpM in Ghanaian SMEs. This proves that effective information sharing and decision support promote SMEs' performance in Ghana. Unlike previous practices of unilateral decision-making by SME owners, current practices stimulate multiple party engagements [14,24]. Consequently, SMEs in Ghana can promote higher performance with a strategically planned supply chain integration with parties like suppliers and distributors. Further, it was witnessed that InC mediates the positive relationship between SupC and FpM in Ghanaian SMEs. While the strength of the mediation is not strong, it could be explained by the growing innovation diffusion in SMEs in Ghana [21]. On the other hand, it could also mean

that the specific SupC activities of SMEs in Ghana focuses less on innovation capability building. Therefore, SMEs in Ghana can boost productivity with the optimal design of SupC activities, which thrives on innovation. Likewise, SMEs can invest in new technologies, methods, and processes to maximize the effect of SupC on FpM. Finally, ShP is found to mediate the positive relationship between SupC and InterVC in Ghanaian SMEs. While the initial strength of the relationship between SupC and InterVC is reduced with the introduction of ShP as the mediating variable, it could explain the least involvement of stakeholders in the decision-making process of SMEs in Ghana a decade ago [27]. Therefore, government policies to stimulate positive stakeholder engagements should be initiated to drive the development agenda of the country through SMEs.

## 6. Conclusions

The study assesses the supply chain integration, interfirm value co-creation, and firm performance nexus in Ghanaian SMEs using the SEM model. The mediating effects of innovation capability and stakeholder pressure are also estimated.

Interfirm value co-creation (InterVC) positively affects SMEs performance (FpM) in Ghana. This suggests that SMEs are more opened to collaboration compared to previous years. Practically, this could be linked to the educational background of the current breed of SME CEOs in the country. InterVC positively affects supply chain integration (SupC) in Ghanaian SMEs. Hypothetically, these activities share similarities, although they are different. The similarity could explain this relationship. Further, the desire of SMEs to save costs, provide efficient services, and improve productivity also accounts for this relationship. Consequently, the current competitive nature of the Ghanaian SMEs industry can be successfully maneuvered with the optimal design of InterVC and SupC activities. On the other hand, InterVC negatively affects innovation capability (InC) in Ghanaian SMEs. While this is not ideal, it also suggests the stage of InterVC practice in Ghanaian SMEs. Further, this negative relationship could be because of the low innovation drive of Ghanaian SMEs. Therefore, InterVC activities should be designed to stimulate innovation within the SMEs.

The pressure exerted by various stakeholders (ShP) positively affects SupC in Ghanaian SMEs. This is because, like any other decision, largely stakeholders seeking optimal usage of organizational resources could drive the engagement of SMEs in SupC activities. This outcome signifies the age of stakeholder engagement in Ghanaian SMEs, unlike in previous years. Therefore, stakeholder engagements in the Ghanaian business environment should be encouraged to drive other significant decisions like the sustainable use of resources and corporate social responsibility. Similarly, ShP positively affects the performance (FpM) of Ghanaian SMEs. Although stakeholders primarily care about the financial welfare of businesses, the non-financial aspect of businesses can also be significantly affected by ShP. Thus, both internal and external stakeholders are critical to the success of SMEs in Ghana. Consequently, these engagements should be prioritized to create a suitable SME ecosystem.

Innovation capability (InC), which involves the use of modern technologies, new methods, new equipment, and procedures, affects SupC in Ghanaian SMEs. This is because most SupC activities thrive on innovation and improved processes. Thus, the success or the extent of SupC activities within SMEs depends on the state of innovation. Ghanaian SMEs should invest in innovation to maximize the benefits of SupC. Therefore, the newly instituted government policies to stimulate innovation diffusion in Ghanaian SMEs are timely for the promotion of this agenda. Again, InC positively affects FpM in Ghanaian SMEs. Improved processes, machines, and equipment drive firm performance. Therefore, the state of technology development, the average level of education, and the age range of Ghanaian SME CEOs provide the perfect platform for learning and diffusion of modern technologies to improve productivity.

Supply chain integration (SupC) positively affects SMEs performance (FpM) in Ghana. This is explained by the fact that SMEs seek avenues to reduce cost while improving

efficiency. The product and services of Ghanaian SMEs can be improved significantly with SupC. Therefore, SMEs can create value, reduce lead-time, and maximize profitability with the practice of SupC. Further, InC mediates the positive relationship between SupC and FpM in Ghanaian SMEs. Nonetheless, this mediation is relatively weak because of the least innovation diffusion in Ghanaian SMEs. Thus, this relationship can be strengthened with investments in new methods, equipment, and processes. Finally, ShP mediates the positive relationship between SupC and InterVC in Ghanaian SMEs. Therefore, stakeholders should be considered integral to the decision-making process of Ghanaian SMEs. This should replace the practice of unilateral decisions by most SME owners in the country. While this study contributes significantly to the literature on supply chain integration, interfirm value co-creation, and firm performance in Ghana, future studies should consider the innovation capabilities of Ghanaian SMEs and the state of stakeholder involvement in SMEs decision making.

## 7. Limitations

The study uses survey responses from only registered members of the Association of Ghanaian Industries. Although this streamlines the study by focusing on formalized SMEs in the country, the non-members' inclusion presents a challenge. Therefore, future studies can focus on non-members to compare the findings from our study. Again, future studies can explore the effect of foreign direct investment on interfirm value co-creation. This is useful because of the increasing number of foreign-owned SMEs in Ghana and the perceived unfair competition from these businesses. The outcome of these studies could trigger policy implications on how to promote collaboration between domestic and foreign-owned businesses.

**Author Contributions:** Author H.T. provided supervision guidance. S.K.O. conceptualized and designed the study. Author C.P.K.C. provided statistical and discussion support. Author I.A.M. provided statistical and modeling support, and author R.K.B. provided proofreading support and referencing support. All authors have read and agreed to the published version of the manuscript.

**Funding:** The National Social Science Foundation of China (14BGL024): Research on the open innovation mechanism and promotional policy of small and medium-sized enterprises from the perspective of Network Embeddedness. Senior talent project of Jiangsu University (08JDG055): Disruptive innovation and construction of SMEs dynamic competitive advantage.

**Informed Consent Statement:** Written informed consent has been obtained from the patient(s) to publish this paper if applicable.

**Conflicts of Interest:** The authors declare no conflict of interest.

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
