# Peer review of "Supply Chain Integration, Interfirm Value Co-Creation and Firm Performance Nexus in Ghanaian SMEs: Mediating Roles of Stakeholder Pressure and Innovation Capability"

_sustainability, doi:10.3390/su13042351_

Round 1

Reviewer 1 Report

The topic of the paper is interesting and up-to-date.

The literature review should be longer and based on more peer-review positions.

Also, the position in literature is not always described according to Journals requirement.

It's worth to add limitation of the paper.

Please add the goal of the paper in the introduction and abstract.

Describe in the methodology about the completeness of the questionnaire. Were they all questionnaires complete? What about non-complete questionnaires? Has it impacted on results?

Author Response

Response to Reviewer 1.

The topic of the paper is interesting and up-to-date.

Response: Thank you for your time, recommendations, and interest in our manuscript. We consider your input useful in improving the quality of the manuscript.

The literature review should be longer and based on more peer-review positions.

Also, the position in literature is not always described according to Journals requirement.

It's worth to add limitation of the paper.

Response: We apologize for not presenting the literature in this format. We have rewritten the entre literature review section to make it more peer-reviewed focus. Accordingly, we have added a limitation and implication for future studies section to the manuscript.

Please add the goal of the paper in the introduction and abstract.

 Response: We apologize for not stating clearly the goal of the study. We have accordingly stated this in the abstract and the introduction of the revised manuscript.

Describe in the methodology about the completeness of the questionnaire. Were they all questionnaires complete? What about non-complete questionnaires? Has it impacted on results?

Response: We have provided that the response rate is as a result of incomplete response and unanswered questionnaire. Further, we have provided the reasons for this situation in the revised manuscript.

Reviewer 2 Report

Dear authors

First of all, it was a pleasure to read your manuscript, a very interesting topic a contemporary also.

Some minor issues:

  • On the abstract "Ghana are unable to fully benefit from these strategies because of the fear of losing trade secrets, inadequate resources, and managerial inefficiencies.", this sentence is not clear to me. Please re-write it in a more clear way.
  • Still on the abstract "To justify the need for these strategies, we employ a structural equation model (SEM)", SEM method is not used to justify strategies choices or adoption, but to analyse influences, impacts, relationships. So I advise the authors to correct that sentence or to explain it more wisely.

Some issues that must be improved:

  • On the section denominated by: "Proposition development" the authors should correct. Because the authors presented a Hypothesis developement, that should be suported by the literature.Each hypothesis must be supported in the literature.
  • On the section denominated by : "Measurement of constructs" the authors should explain in more detail form what were the questions referent to each variable.

Author Response

Response to Reviewer 2.

First of all, it was a pleasure to read your manuscript, a very interesting topic a contemporary also.

Response: Thank you for your time and recommendations for improving the quality of our research. We consider your input significant to improving the readability of the paper.

Some minor issues:

  • On the abstract "Ghana are unable to fully benefit from these strategies because of the fear of losing trade secrets, inadequate resources, and managerial inefficiencies.", this sentence is not clear to me. Please re-write it in a more clear way.

Response: This statement has been deleted and revised to provide clarity.

  • Still on the abstract "To justify the need for these strategies, we employ a structural equation model (SEM)", SEM method is not used to justify strategies choices or adoption, but to analyse influences, impacts, relationships. So, I advise the authors to correct that sentence or to explain it more wisely.

Response: We have revised this section accordingly.

Some issues that must be improved:

  • On the section denominated by: "Proposition development" the authors should correct. Because the authors presented a Hypothesis developement, that should be suported by the literature.Each hypothesis must be supported in the literature.

Response: We have rewritten this section and included justification for the proposed research hypothesis.

  • On the section denominated by : "Measurement of constructs" the authors should explain in more detail form what were the questions referent to each variable.

Response: This section has been revised to provide further explanation of the type of questions asked in the survey.

Round 2

Reviewer 1 Report

Authors have implemented all my remarks.

Author Response

Response to Reviewer 1.

Authors have implemented all my remarks.

Response: Thank for the positive evaluation of our manuscript. We have seen significant improvement in the quality of the manuscript after implementing your recommendations. Further, we have proofread the article.

Reviewer 2 Report

First of all, the authors have improved a lot the paper in comparison to the previous version.

Some minor issues:

  • "Therefore, we speculate hypotheses 3a and 3b to estimate the influence of supply chain integration and interfirm value co-creation on the innovation capability of Ghanaian SMEs." The term speculate is not suitable for a scientific publication.
  • The authors should do an English proof-reading, in order to improve the quality of English.

Author Response

Response to Reviewer 2.

First of all, the authors have improved a lot the paper in comparison to the previous version.

Response: Thank you for your positive evaluation.

Some minor issues:

  • "Therefore, we speculate hypotheses 3a and 3b to estimate the influence of supply chain integration and interfirm value co-creation on the innovation capability of Ghanaian SMEs." The term speculate is not suitable for a scientific publication.

Response: We have corrected this statement and all others.

  • The authors should do an English proof-reading, in order to improve the quality of English.

Response: We have proofread the manuscript using Grammarly.